# Early crustal processes revealed by the ejection site of the oldest martian meteorite

A. Lagain [1✉], S. Bouley[2,3], B. Zanda[3,4], K. Miljković [1], A. Rajšić[1], D. Baratoux[5,6], V. Payré[7], L. S. Doucet [8], N. E. Timms[1,9], R. Hewins [4,10], G. K. Benedix [1,11,12], V. Malarewic[2,4], K. Servis [1,13] & P. A. Bland[1]

The formation and differentiation of the crust of Mars in the first tens of millions of years after its accretion can only be deciphered from incredibly limited records. The martian breccia NWA 7034 and its paired stones is one of them. This meteorite contains the oldest martian igneous material ever dated: ~4.5 Ga old. However, its source and geological context have so far remained unknown. Here, we show that the meteorite was ejected 5–10 Ma ago from the north-east of the Terra Cimmeria—Sirenum province, in the southern hemisphere of Mars. More specifically, the breccia belongs to the ejecta deposits of the Khujirt crater formed 1.5 Ga ago, and it was ejected as a result of the formation of the Karratha crater 5–10 Ma ago. Our findings demonstrate that the Terra Cimmeria—Sirenum province is a relic of the differentiated primordial martian crust, formed shortly after the accretion of the planet, and that it constitutes a unique record of early crustal processes. This province is an ideal landing site for future missions aiming to unravel the first tens of millions of years of the history of Mars and, by extension, of all terrestrial planets, including the Earth.

[1] Space Science and Technology Centre, School of Earth and Planetary Science, Curtin University, Perth, WA, Australia. [2] Université Paris-Saclay, CNRS, GEOPS, 91405 Orsay, France. [3] IMCCE, Observatoire de Paris, 77 avenue Denfert-Rochereau, 75005 Paris, France. [4] Institut de Minéralogie, de Physique des Matériaux et de Cosmochimie (IMPMC), Muséum national d'Histoire naturelle, Sorbonne Université et CNRS, 75005 Paris, France. [5] Géosciences Environnement Toulouse, University of Toulouse, CNRS and IRD, Toulouse 31400, France. [6] Université Félix Houphouët-Boigny, Abidjan, Côte d'Ivoire. [7] Department of Astronomy and Planetary Science, Northern Arizona University, Flagstaff, AZ, USA. [8] Earth Dynamics Research Group, TIGeR, School of Earth and Planetary Sciences, Curtin University, Perth, WA, Australia. [9] The Institute for Geoscience Research (TIGeR), Curtin University, Perth 6845 WA, Australia. [10] EPS, Rutgers University, Piscataway, NJ 08854, USA. [11] Department of Earth and Planetary Sciences, Western Australian Museum, Perth, WA, Australia. [12] Planetary Sciences Institute, Tucson, AZ, USA. [13] Pawsey Supercomputing Centre, CSIRO, Kensington, WA, Australia. ✉email: anthony.lagain@gmail.com

The geological record of the formation and differentiation of our planet has been destroyed by its subsequent evolution, but extremely rare clues may be obtained from other terrestrial planets. Mars provides a unique and accessible example of an early evolutionary path corresponding to that, inaccessible, of our own world. We can investigate it with spacecraft, and samples are available for in-depth analysis on Earth in the form of martian meteorites. So far, the only available martian samples that appear to have recorded the early conditions and the evolution of the planet until the present time are Northwest Africa (NWA) 7034 and its paired stones. They are the most diverse martian meteorites in terms of composition, containing a variety of igneous, sedimentary, and impact melt clasts, including the most evolved and oldest igneous clasts and zircons (4.44–4.48 Ga old[1–10], grey bars in Fig. 1). These evolved clasts are derived from a variety of magmas (monzonitic or mugearitic) and probably formed by re-melting of the primary martian crust either at various depths in the presence of volatiles or by differentiation of large impact melt sheets[9,10]. The abundance of trace elements reported in the old zircon population[5] indicates a variability of U/Yb ratios, which suggest different types of source rocks and processes for the genesis of these magmas. These old evolved clasts have most likely been excavated by an impact event during the Early Amazonian period, ~1.5 Ga ago[3,5,8,11–13] (green bars in Fig. 1), before being lithified[12] and ejected ~ 5 Ma ago[12,13] (Methods). Hence, this regolith breccia provides evidence for the formation of evolved crustal material 4.48 Ga ago on Mars[1–10,14,15], and contains clues to the early environment and evolution of Mars.

However, the source region of this unique meteorite and its geological context have so far remained unknown, and with it, a region where the earliest geological records of the planet[2,3] are exposed on the surface. Knowing this source region would provide insights into early Mars geological history and crustal extraction[2,3]. This source region may therefore become a high-priority target for detailed orbital analyses and in-situ exploration[16].

Following a hypervelocity impact, ejecta materials moving faster than the escape velocity (5 km/s)[17] may get through the martian atmosphere and continue their course into interplanetary space to become martian meteorites. Slower debris fall back on the surface in a radial pattern or ray around the primary crater,

forming secondary craters. The presence of 100 meter-size secondaries attests to the freshness of their associated primary craters[18]. Using the size and spatial distribution of more than 90 million impact craters >50 m (both primaries and secondaries) detected using a Crater Detection Algorithm (CDA)[18–20] on the whole surface of Mars from the global Context Camera (CTX) mosaic[21], a previous work[18] identified ray systems of secondary craters <150 m associated with 19 large primary craters. For each of them, a formation model age was measured using small craters superposed on their ejecta blanket, and 18 were found younger than 10 Ma old. The analysis of the size frequency distribution of these 18 young crater candidates revealed that those larger than 7 km (i.e., 17 out of 18) align with the predicted number and size of craters accumulated on the whole surface of Mars over the last 8.2 ± 2 Ma[22]. Hence, those impact craters were found to constitute the complete crater population >7 km in diameter formed on Mars over the last ~10 Ma, potentially responsible for the ejection of martian meteorites[18]. One of these craters, Tooting, has already been recognized as the most likely ejection site of the depleted olivine-phyric shergottites launched 1.1 Ma ago, located on the Tharsis volcanic province[18].

Past studies that have proposed parent terrains for the unique NWA 7034 regolith breccia all agree that it must come from the heavily cratered southern Noachian highlands of Mars[7,11,13,23–26] based on their notable geophysical and geochemical affinities, such as the elevated concentrations in potassium (K), thorium (Th), and iron (Fe) in the meteorite[8,9,23–26], the ages of the oldest minerals found in the breccia[1,5,8–10], its unique magnetic signature[7], and its visible-infrared reflectance spectra[24].

In this work, we search for the most likely site of ejection of the regolith breccia by using four criteria based on its geochemical and geophysical properties as well as its geochronological records (Methods), which we compare to potential sites based on their known properties and geological context: (1) high magnetic field intensity and remanent magnetization at the surface from up-to-date orbital dataset[27] (Methods, Fig. 2b, c); (2) high elemental Th and K concentrations[28,29] (Methods, Fig. 2d, e) of the areas surrounding each crater candidate; (3) superposition on a Noachian geological unit[30] and (4) connection with material from an Early Amazonian impact. We show that only one crater candidate characteristics match with the meteorite properties. The oldest clasts of NWA 7034 and paired stones were excavated ~1.5 Ga

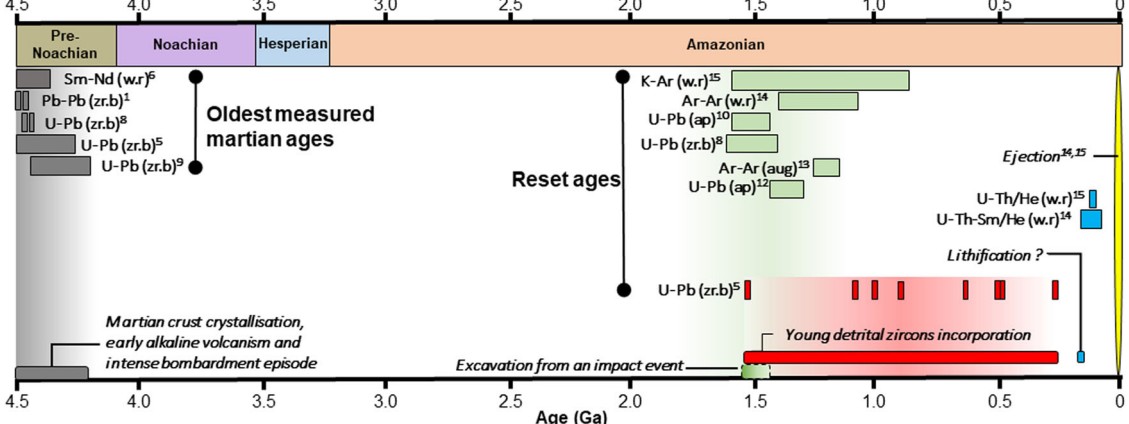

**Fig. 1 Summary of NWA 7034 and paired stone radiometric ages, and chronology of major events experienced by the breccia.** Dates from each study are reported in Supplementary Data 1. The ejection event is constrained from $^{22}Ne/^{21}Ne$ cosmic ray exposure ages[12, 13]. Whole rock (w.r), zircon and or baddeleyite (zr.b), apatite (ap), and augite (aug) on which ages have measured are also mentioned. The different chronometers used in these studies are reported (Sm-Nd, Pb-Pb, U-Pb, K-Ar, U-Th/He, U-Th-Sm/He). Note that green, red and blue boxes correspond to resetting ages of the noted chronometer. Reset ages in green are widely interpreted as the disruption induced by an impact-derived heating event that has excavated the oldest components of the breccia ~1.5 ago[9, 11, 23, 26], although its precise age is still unconstrained due to the wide range of isotopic dates reported in the literature.

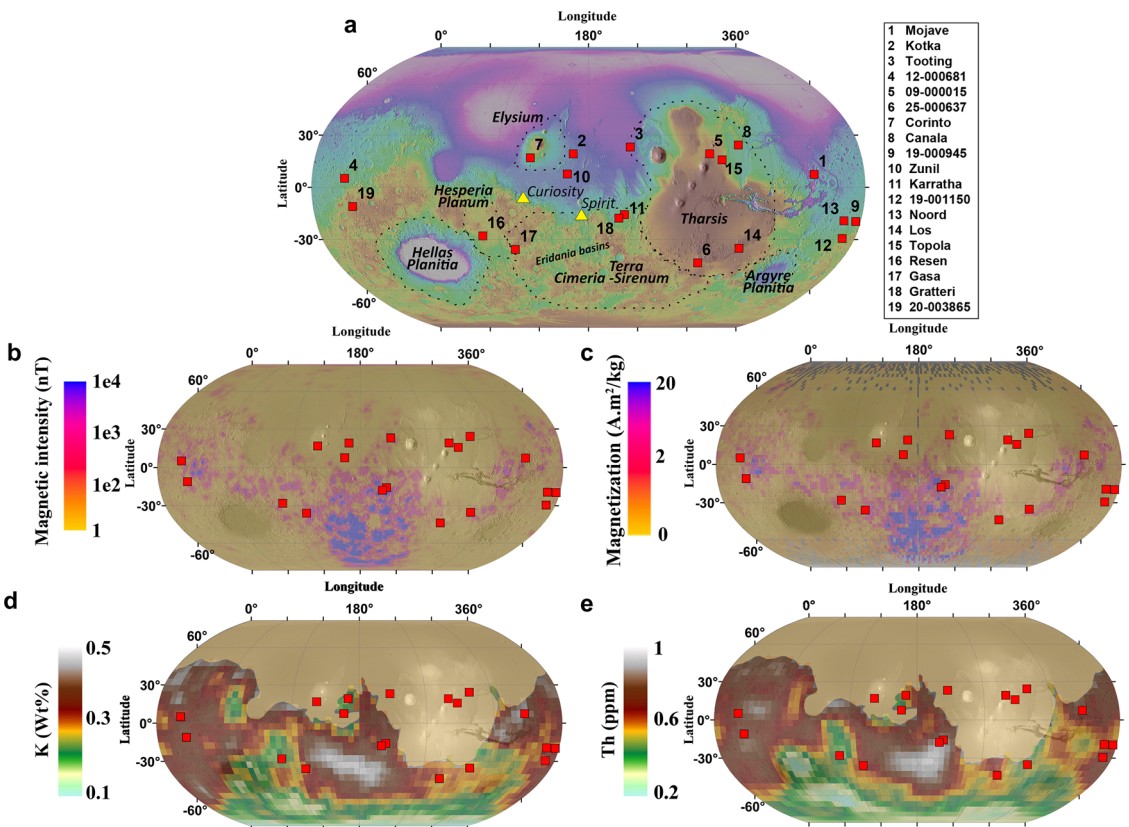

**Fig. 2 Distribution of the most likely crater sources for martian meteorites. a** Global context of the 19 crater candidates[18] (Supplementary Table 1) and location of provinces and rovers (yellow triangles) referred to in the present study. Background: Mars Orbiter Laser Altimeter (MOLA) shaded relief (https://astrogeology.usgs.gov/search/map/Mars/GlobalSurveyor/MOLA/Mars_MGS_MOLA_DEM_mosaic_global_463m). **b**, **c**: Magnetic field intensity and remanent magnetization at the surface from ref. [27]. **d**, **e** Potassium and Thorium concentration at the surface from ref. [28,30]. Beige area corresponds to discarded provinces due to the weathered basaltic surface contribution (Methods).

ago by an impact that has formed a 40 km crater. The ejecta material of this crater were subsequently ejected by a second impact a few million years ago, which led to the formation of a 10 km crater. The geologic context of the ejection site is consistent with one of the oldest province of Mars, a relic of the differentiated primordial martian crust. This region constitutes a unique record of the first tens of millions of years of the history of Mars.

## Results

Only nine craters of the previously identified large and recent primary craters[18] are located on Noachian highlands terrains[30] (Fig. 2a and Supplementary Table 1). A high temperature event is required to account for the metamorphic changes experienced by NWA7034 and paired stones 1.5 Ga ago[3,8–10,12–14] (Methods). None of these nine craters are associated with Amazonian volcanic terrains[30], thus making implausible a volcanic origin for the 1.5 Ga event that reset radiochronometers in the apatites, feldspars and zircons within the breccia[31]. On the other hand, two of these nine craters, Karratha and Gasa, respectively labeled 11 and 17 in Fig. 2a (Supplementary Table 1) are superposed on the crater floor or an ejecta blanket of an Amazonian impact crater (younger than ~3.2 Ga old). Such an Amazonian crater can account for the 1.5 Ga event that led to the excavation of the Noachian basement, its brecciation and resetting of the radiochronometers, and the lithification in the ejecta deposits.

While Gasa crater has been noticed in previous studies due to its extended rays visible on thermal imagery[32,33], Karratha crater is devoid of such thermally visible rays, and, to our knowledge, has never been reported before as a young primary crater. Gasa crater is located within Cilaos crater (20 km). A model age of $572 \pm 110$ Ma was estimated from crater counts for the Cilaos impact event (Methods and Supplementary Fig. 4). Cilaos crater is hence too young to be associated with the 1.5 Ga resetting event. Moreover, magnetic signatures and elemental abundances of K and Th reported in the region surrounding Gasa crater are lower compared to those associated with Karratha (Supplementary Fig. 3 and Supplementary Table 1).

Secondary craters from Karratha extend over more than 350 km (Supplementary Fig. 5). It is located within a highly degraded 25 km impact structure, Dampier crater, most likely Noachian in age (Fig. 3). This old crater is filled by the ejecta blanket of a nearby impact crater (Khujirt, D = 40 km), located about 30 km away (rim to rim) in the south-west of Karratha. Karratha crater is superposed on this ejecta material, whose formation occurred during the Early Amazonian period, between $1.25^{+0.38}_{-0.32}$ and $1.87^{+0.73}_{-0.65}$ Ga (Methods and Supplementary Fig. 4). An estimate for the thickness of the Khujirt ejecta blanket from scaling laws in ref. [34] gives 60 m where Karratha is located. The maximum depth of debris reaching the martian escape velocity following a 350 m asteroid impact forming a 10 km size crater is ~50 m (ref. [17]). Hence, the large majority of the ejected debris capable of escaping martian gravity following the formation of Karratha are from the Khujirt ejecta blanket, not from the underlying material related to the Dampier crater (Fig. 3).

Moreover, orbital datasets indicate that the Karratha crater is associated with particularly high concentrations of K and Th and

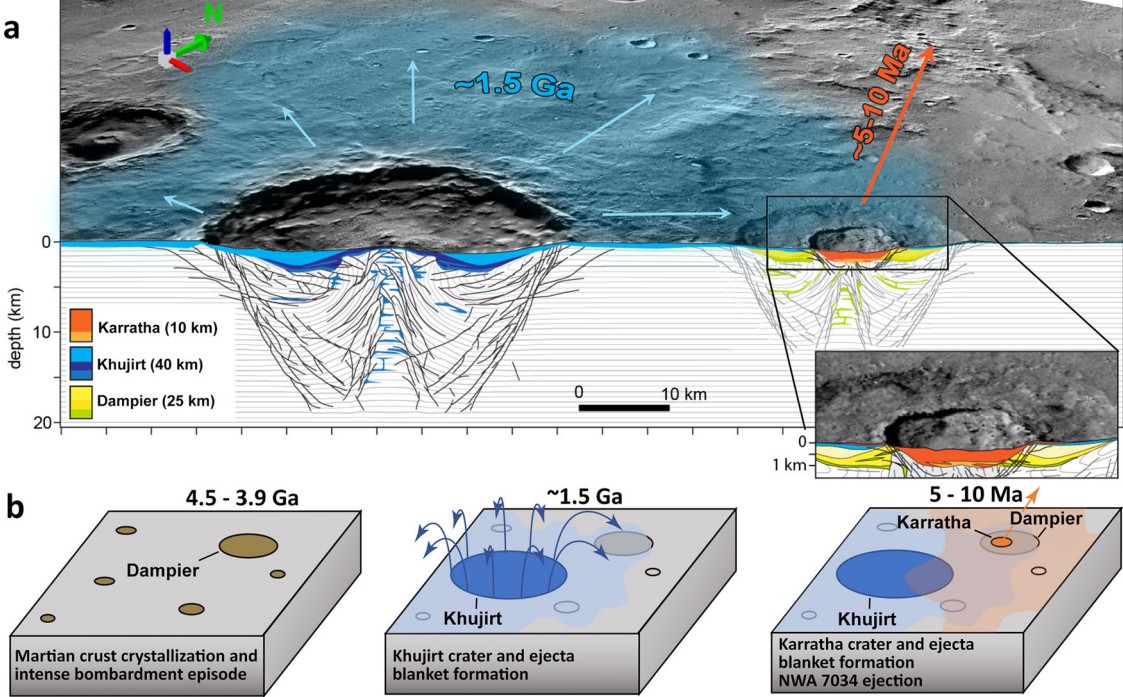

**Fig. 3 The NWA 7034 launch site geological context. a** Perspective view (Context Camera mosaic[21]) and cross-section through Karratha along a SW-NE axis (Supplementary Fig. 5b) as interpreted from numerical modelling simulations. Shades of colors denote impactites (impact melt and in situ breccia; ejecta and fall-back breccia). **b** Schematic of chronological events experienced by the host terrain of the regolith breccia.

elevated magnetic intensities, when compared to all other craters, although these values are lower than those measured in NWA 7034 and paired meteorites (Supplementary Fig. 3). The Karratha crater appears therefore a good candidate for the launch site of NWA 7034, if the formation condition of Khujirt was intense enough to account for the ~1.5 Ga resetting ages measured in augite, apatite, and zircon (Methods). We modeled the temperature of ejecta fragments associated with the formation of the Khujirt crater using the iSALE shock physics code[35–37] (Methods). The simulation indicates that the ejecta fragment experienced a temperature up to 1000 °C where Karratha is located (Supplementary Figs. 6 and 7), compatible with the temperature required to reset the U-Pb in apatites, the Ar-Ar in augites and to disturb the metamict zircons in the breccia[3,5,8,10–15] (500–800 °C, see Methods). With the exception of a shock twin observed in a zircon contained in NWA 7034[38], the paucity of shock deformation above 10 GPa observed in the breccia[39] is also consistent with an origin as poorly consolidated ejecta material for the regolith breccia.

The four criteria used to locate the crater source of the regolith breccia thus allow nailing down the young crater candidate population to a unique solution. The geological context of Karratha matches the chronology, the lithology, and the magnetic and elemental signatures of the NWA 7034 meteorite group. We conclude that the ejecta blanket on which Karratha is superposed, is associated with the Early Amazonian impact event (Khujirt, in green in Fig. 1) that has excavated the oldest zircons and clasts present in the breccia from the Noachian southern highlands (in grey in Fig. 1). The impact that has formed Karratha crater has subsequently ejected material of the Khujirt crater a few million years ago (in yellow in Fig. 1), making the Karratha impact crater the source of NWA 7034 and paired stones (Fig. 3).

**Implications for the early crust extraction**. The bedrock of the Khujirt crater, located in the north-east of the Terra Cimeria—

Sirenum region (here and after noted TCTS) is believed to be composed of basaltic and more evolved lithologies such as those represented by the monzonitic and noritic clasts, containing the concordant 4.4 Ga zircons[1–5,8–10]. The TCTS province is located between Hesperia Planum and the Tharsis bulge (Fig. 2a), and characterized by the highest concentrations in K (>0.35 wt.%) and Th (>0.35 ppm) measured on Mars from the orbit[28,29] (Fig. 2b, c). TCTS is the only highlands region where the high concentrations of both elements are correlated[29]. Furthermore, it presents the highest magnetic field anomaly (> 5000 nT) and the strongest remanent magnetization (>5 A.m²·kg⁻¹) on Mars[27] (Fig. 2d, e).

By removing the contribution of the largest impact basins and volcanoes from the gravity field and topography, it has been found that the TCTS province is characterized by the highest crustal thickness of the planet, i.e., >50 km[40]. Since this terrain is overprinted by the Hellas and Argyre basins' ejecta materials, formed earlier than ~4.1 Ga, this region is likely a relic of the most ancient crust[40], which is confirmed by the location of the NWA 7034 ejection site. The compositional effect of the crust on the magnetic intensity[41] suggests that TCTS remained largely unaffected by demagnetizing processes since the pre-Noachian, and that surface material has not been mixed with the surrounding Noachian regolith. This is consistent with a thick crustal block[40], whose distinct formation and evolution possibly reflect the first stage of differentiation occurring very early in the history of the planet. The geological stability of this region makes it unlikely that Amazonian and Hesperian hydrothermal processes have contributed to forming magnetite to be the dominant magnetization carrier in this highly magnetized region of the martian crust[42,43]. Instead, analysis of magnetic signatures, K and Th enrichment, and depth of large impact craters in Eridania basins within the TCTS region suggest non-magmatic long-lived hydrothermal systems, heat-driven by radiogenic elements with half-lives of billions of years[44] (e.g., $^{232}U$ and $^{40}K$), that might have significantly contributed to the observed

crustal magnetic field throughout the pre-Noachian and Noachian eras[42–44]. Early hydrothermal circulation in this province would potentially have sustained life-compatible environment for a long period of time[3,44].

The TCTS region covers about 10% of the planet and has been interpreted as a crustal block characterized by a geochemically evolved component. Evolved rocks have been observed and analyzed on the ground in Gale and Gusev craters by Curiosity and Spirit, respectively, in the immediate vicinity of this province, (Fig. 2a). Those igneous rocks show felsic alkaline and sub-alkaline compositions[45–49] that might be explained by fractional crystallization[47,48], and may indicate the presence of a differentiated crust early in the history of the planet[46].

The U-Pb and Pb-Pb isotopic compositions in monzonitic clasts in the meteorite suggest the existence of an isotopically enriched (relative to the martian mantle) and differentiated crust on Mars[1–5,14,50] that was extracted before 4.547 Ga[1]. According to the Hf isotopic signature of 4.43 Ga zircons recovered from NWA 7034, it has been proposed that a magma ocean crystallized within the first 20 Ma after the accretion of the planet[1]. The initial εHf value of these old zircons and models of early magma ocean crystallisation[51–53] imply an andesitic composition for the early martian crust[1,5,53], although the relationship between U-Pb ages and εHf of the oldest zircons suggest that they cristallized from low $^{176}Lu/^{177}Hf$ magmas potentially of basaltic affinity[5]. Such a crust was then reworked 100 Ma later by impacts[9,10], producing the melts from which the old zircons crystallized. The analyses of >3.8 Ga evolved rocks in Gale crater[46], the inversion of the martian gravity field (constrained by petrological data that support the existence of light evolved crustal components -less dense than basalt- in the southern highlands)[54] and, finally, seismic data from the *Insight* mission (indicating that the rate of P wave against S wave is compatible with basaltic to andesitic crustal materials[55] and suggesting that the crust density is <3100 g.cm$^{-3}$, so lower if only composed by basaltic rocks[56]), all point out to the presence of highly ancient evolved crustal components in TCTS.

If andesitic in composition, the crust could either be secondary (reworking of the primordial crust) or primordial. If primordial, basaltic to andesitic melts that originated from a deep mantle source might have crystallized[57]. However, isobaric partial melting experiments[58] and adiabatic ascent of primitive mantle compositions[48] argue against the formation of andesitic magmas under such a scenario. If secondary, the primordial crust would have been extracted from the magma ocean extremely early, before 4.547 Ga ago, i.e., <20 Ma after solar system formation[1]. Alternatively, if the primordial crust was basaltic in composition, its differentiation and re-melting might have resulted in an evolved crust as observed for the continental crust on Earth. Another possibility is the absence of a global magma ocean, as suggested by ref. [8], where a low-degree of partial melting of a fertile mantle could produce an enriched crust with rare-earth element patterns similar to those observed within the regolith breccia. In the later case, scattered magma oceans could have occurred in locations different from the source terrain of the breccia. Differentiated primordial crustal blocks of the planet such as that in TSTC would have been formed by simple partial melting shortly after the accretion of the planet.

In any case, we suggest that clasts contained in the regolith breccia are representative of the TCTS province, making this region a relic of the early crustal processes on Mars, and thus, a region of high interest for future missions. The study of TCTS would help us unravel the conditions of formation and the first evolution stage of Mars, and by extension of all terrestrial planets, considering the fact that, in light of these findings, early crustal processes appear uniquely preserved and accessible on Mars. The flanks and central peaks of large and preserved craters within this region might constitute outcrops of high interest, containing the missing geological clues to the early-stage evolution of Mars.

## Methods

**NWA 7034 and pair characteristics.** The diversity of clasts contained in the breccia makes this meteorite one of the martian samples with the most complex history, recording multiple events, from the crystallization of the martian primary crust to the ejection of the rock. The crystal clasts contained in the breccia include low-Ca pyroxene, augite, plagioclase and alkali feldspar, with a near absence of olivine. The fine-grained matrix contains pyroxene, plagioclase, iron oxides, Cl-apatite, chromite and pyrite[5,8,11,26,59]. A vitrophyre melt clast containing the highest Ni abundance ever measured in any martian meteorite or igneous rock (1020 ppm) has also been reported[50], suggesting contamination by a chondritic impactor, also seen in all lithic clast types except orthopyroxenite[9,26]. Basaltic clasts and bulk matrix compositions of this meteorite have been found to be analogous to igneous rocks analyzed by the Spirit rover within Gusev crater as well as the average martian crust composition determined by the Gamma-Ray Spectrometer (GRS) aboard the Mars Odyssey spacecraft[53]. However, some evolved clasts exhibiting trachyandesitic, basaltic, and Fe-, Ti-, and P- rich (FTP) lithologies constitute rock types similar to those analyzed by the Curiosity rover in Gale crater[45,46,48,49].

This meteorite exhibits unique characteristics, including one of the highest concentrations in potassium and thorium ever measured in a martian meteorite (Supplementary Fig. 1a). Coupled with isotopic analysis ($^{147}Sm/^{144}Nd$ and $^{176}Lu/^{177}Hf$), this suggests that the mantle source of NWA 7034 is isotopically and chemically distinct from the other martian meteorites mantle sources[2–10]. The unique magnetic mineralogy of the breccia makes NWA 7034 the most magnetized martian meteorite[7] with remanent magnetization (20–60 A/m) one order of magnitude higher than any other martian meteorites (Supplementary Fig. 1b). Metamict zircons[8,10] apatites[15], augite[11] and alkali-feldspars in leucocratic clasts[12] age measurements suggest the occurrence of one single metamorphic event ~1.5 Ga ago (in green in Fig. 1). This event has been interpreted either as a volcanic one[31] inducing protracted metamorphism or as an impact that would have excavated the oldest component of the meteorite, with temperatures ranging between 500 °C and 800 °C (refs. [3,9,11,26]). Grain shape and size distributions of the breccia clasts indicate they were likely deposited by impact-ejecta materials under base surge conditions[11]. Moreover, the petrographic similarity of the meteorite with terrestrial suevite[9,11], accretionary dust rims[23]), and the presence of stishovite[60] (a high-pressure polymorph of $SiO_2$) all favor an impact origin for the resetting of the U-Pb in augite and alkali-feldspars and the disturbance of the metamict zircons in the breccia[8,10,11]. Suevite is described as a polymict breccia containing lithic and mineral fragments that exhibit a variety of shock metamorphism stages[61,62]. It usually composes the upper layer of the crater cavity and the proximal ejecta layer[61]. Based on the grain size and shape, previous work[9] suggested that the meteorite is representative of a proximal ejecta blanket deposited in a pyroclastic flow regime, consistent with conclusions from ref. [11], according to which the accretionary dust rims seen in the breccia have been formed under base surge conditions following an impact event, ~1.5 Ga ago.

Eight young detrital zircons were recently discovered[5] in NWA 7533 with ages from 1548.0 ± 8.8 Ma to 299.5 ± 0.6 Ma (in red in Fig. 1). Analysis of their isotopic composition indicates a common mantle source, sampled by deep-seated magmatic activity, possibly representative of Tharsis or Elysium volcanic provinces[5] These zircons were likely transported by eolian processes to the source region of the breccia[5]. The oldest grain being 1548 Ma old[5], this zircon population is consistent with an excavation ~1.5 Ga ago, possibly triggered by an impact. The breccia lithification would have occurred subsequently, 225 Ma ago[12] (in blue in Fig. 1), while the ejection took place ~5 Ma ago[12,13] (in yellow in Fig. 1).

**Constraints from orbital dataset.** We compare the abundance of K and Th as well as the magnetic field intensity and the magnetization of the surface of Mars derived from orbital measurements at the immediate vicinity of each crater candidate with those of the breccia. The concentrations in K and Th are from the Gamma-Ray Spectrometer (GRS) aboard the Mars Global Surveyor (MGS)[28,29]. Although the spatial resolution of the data is low (5°x5°, corresponding to ~296 km at the equator), this dataset offers a consistent method to compare the concentration of both elements qualitatively between several regions where the crater candidates are located[63]. We report here both the pixel value of the concentration of K and Th at the crater location (Supplementary Table 1) and the bilinear interpolation computed with a radius of 296 km around each crater centroid (Supplementary Table 2).

A recent model of the crustal magnetic field of Mars has been computed using the Mars Atmosphere and Volatile EvolutioN (MAVEN) magnetometer[27]. It is the highest resolution model of the martian magnetic field at the surface (spatial resolution: ~100 km/px, magnetic field resolution: <1 nT), allowing the identification of small-scale features associated with geological signatures (Fig. 2b). From this model, the authors also derived the equivalent magnetization distribution (Fig. 2c). Because magnetic fields and magnetization spatial variation exist at a small scale (smaller than the spatial resolution of the model and thus

lower than the magnetization currently detectable from orbit[25], we report for each crater candidate two values of the magnetic field intensity and equivalent magnetization: (1) the pixel value associated with the centroid of the crater (Supplementary Table 1 and Supplementary Fig. 3) and (2) a bilinear interpolation computed from a 100 km buffer (resolution of the model) computed around each crater centroid (Supplementary Table 2).

In order to distinguish crater candidates associated with a host terrain exhibiting relatively high values of the four orbital datasets considered here, we compute the first standard deviation from the kernel density distribution of the data. Values higher than +1σ are considered relatively high (Supplementary Fig. 2) and used to discriminate areas with elemental concentration and magnetic properties that might be consistent with the exceptional characteristics of the breccia qualitatively (Supplementary Fig. 3). Enrichment in K and Th in the meteorite being related to magmatic processes, we chose to discard the northern lowlands, and more generally areas above the dichotomy, to compute the distribution of the K and Th concentration due to the weathered basaltic surface contribution[64] that may not represent the bedrock chemistry on those provinces.

Supplementary Fig. 3 presents the chemical and magnetic signatures for each crater candidate (in pixel values). The high values range associated with each dataset are represented by the white area and impact craters are color-coded as followed: craters located on Noachian geological unit[30] and superposed on the material of an Amazonian impact crater (cavity or ejecta) are in green. Those that are only located on Noachian material are in orange, and craters superposed on the ejecta of an Amazonian impact crater are in red. Finally, if none of these two criteria are respected, craters are shown in grey.

**Model age derivation of impact events.** Karratha crater is superposed on the ejecta blanket of a 40 km crater (Khujirt, in blue in Fig. 3) that filled the cavity of an older crater, most likely of Noachian age (Dampier, in orange in Fig. 3). Using the Context Camera (CTX) mosaic[21], we mapped impact craters superposed on both the 40 km crater cavity (for D > 100 m) and its ejecta blanket (for D > 500 m) to estimate the age of the material surrounding Karratha, i.e., the ejecta blanket of the 40 km crater (Supplementary Fig. 4c, d). For this, crater mapping is performed by using the CraterTools software[65] and Crater-Size Frequency Distributions (CSFD) are loaded into CraterStats II[66] and fitted with an isochron using a standard chronology model[22]. We derived all model ages using the differential representation[67,68]. Compared to a cumulative plot, the differential representation allows easier recognition of any resurfacing event contribution, the presence of potential overprinting craters formed prior to the ejecta blanket in the population of mapped craters, or secondary craters[67]. Each point in the CSFD is independent of the subsequent larger diameter bins. Like other representations, the binning of the data can lead to biases when the CSFD is fitted with an isochron[69] when a small number of impact craters is used to derive model ages[70]. We solved this statistical disadvantage by using the Poisson probability-density function-fitting technique[69]. This solution allows an exact prediction of the model crater chronology model according to the CSFD considered, whatever the chosen binning technique. The crater count on the ejecta blanket of Khujirt crater leads to a model age of $1.87 \pm^{0.73}_{0.65}$ Ga, consistent with the model age obtained from crater count on the crater cavity $1.25 \pm^{0.38}_{0.32}$ Ga.

Gasa crater is located within the Cilaos impact crater. The ejecta blanket of the latter has been mapped (Supplementary Fig. 4c) and crater counts have been performed using the CTX global mosaic[21] down to ~250 m to estimate the age of the Cilaos impact crater. A middle Amazonian model age[22] (572 ± 110 Ma) is obtained using the same technique and chronology model used for Karratha crater (Supplementary Fig. 4d). Even taking into account larger uncertainties of the crater count method linked to the influence of the terrain rheology on the crater size[71] or potential fluctuation in the impact cratering rate and crater production[67,72–74], the model age derived for the Cilaos impact event is inconsistent with the 1.5 Ga resetting age of some minerals in the meteorite[8,10–12]. In summary, the impact that has formed the ejecta blanket of Khujirt on which Karratha is superposed occurred most likely during the early Amazonian period[20], ~1.5 Ga ago. This impact event is the only one that could match the 1.5 Ga resetting age observed in the meteorite (in green on Fig. 1) as well as the age of the oldest detrital zircon[5] (in red on Fig. 1), consistent with the excavation of the parent rock of NWA 7034.

**Impact crater modeling and ejecta temperature.** We model the shock and post-shock temperature of the ejecta curtain associated with the formation of a ~40 km crater (i.e., Khujirt) using iSALE-2D (refs. [35–37]). (Supplementary Fig. 6 and Supplementary Table 3). For this, we use a 4.5 km diameter impacting the surface at 9.6 km/s (ref. [74]). Impactor and target were modeled using the equation of state for dunite[75] and basalt[76], respectively. The temperature gradient in the upper crust is assumed to be 15 K/km (ref. [77]), which is on the hotter side of present-day Mars and could be appropriate for Mars at 1.5 Ga. We find that ejecta material experienced a heating ranging between 0 and >1000 °C where Karratha was formed (~1.5 crater radii from the rim of Khujirt crater, Supplementary Fig. 7). This is compatible with the temperature required to disturb the Ar-Ar and U-Pb radio-chronometers in the breccia[3,5,8,10–13,15] (500–800 °C). Depth of origin for this ejecta material does not exceed 5 km. We note that the heating of the landing ejecta is directly dependent on the impactor speed. Increasing impact speed and lowering the size of the projectile so that the resulting crater diameter is the same, the mean

temperature range in ejecta will elevates but remains in the observed range. Furthermore, any macro voids within the falling ejecta could significantly elevate the temperature in the falling ejecta[78]. It is therefore difficult to estimate the duration of the shock temperatures necessary for chronometer resetting, as that would depend on a number of parameters such as mineral composition, sample/ejecta fragment size, porosity, heterogeneity, etc. Direct comparison between numerical modelling outcomes and laboratory measurements is not directly applicable, and merits further work.

## Data availability

The data that support the findings of this study are available within the paper and the Supplementary Data file.

## Code availability

The numerical impact crater formation were made using the iSALE shock physics hydrocode. At present, iSALE is not fully open source. Application for use of iSALE can be made via https://isale-code.github.io/. Any recent stable release can be used to reproduce the data presented. We used the IDL 5.2 software (L3Harris geospatial https://www.l3harrisgeospatial.com/Software-Technology/IDL) to run the CraterStats II software available at https://www.geo.fu-berlin.de/en/geol/fachrichtungen/planet/softwarealgorithm, and the ESRI's ArcGIS 10.8.1 software suite (ESRI https://www.esri.com/en-us/arcgis/about-arcgis/overview) and Matlab (https://au.mathworks.com/products/matlab.html) to produce the maps.

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

## Acknowledgements

The authors acknowledge Benoit Langlais for providing guidance and magnetic field data, and Denis Fougerouse for discussions and guidance regarding age measurements used in this study. We gratefully acknowledge the developers of iSALE-2D, including Gareth Collins, Kai Wünnemann, Dirk Elbeshausen, Tom Davison, Boris Ivanov and Jay Melosh. This research was funded by the Australian Research Council grants DP170102972, DP210100336, DP180100661, DE180100584, and FT210100063, Curtin University, the Western Australian Government, and the Australian Government.

## Author contributions

A.L, S.B., and B.Z. conceived the project. A.L. performed the orbital dataset analyses, derived model ages, and investigated the geological context of all crater candidates. K.M. and A.R. performed the impact simulations. A.L., S.B., B.Z., D.B., V.P., L.S.D., and R.H. discussed the results. A.L. drafted the manuscript with contributions and critical feedback from S.B., B.Z., K.M., A.R., D.B., V.P., L.S.D., N.E.T., R.H., G.K.B., V.M., K.S., and P.A.B.

## Competing interests

The authors declare no competing interests.
