## [Peer Review File · Nature Communications]

REVIEWER COMMENTS

Reviewer #1 (Remarks to the Author):

This paper reports an analysis of Mars' s cratering record aimed at identifying the source region of the NWA 7034 martian meteorite and its pairs (i.e. NWA 7533). These meteorites contain minerals and fragments of ancient crust that date back to 4.5 Ga and, as such, provide insights into the earliest crustal evolution on Mars. Thus, a robust identification of the launching site of these samples is important as it may guide future robotic exploration campaigns that aim to sample the primordial crust of Mars.

The authors' analysis suggests that the NWA 7034 meteorite suite was ejected around 5-10 Ma from the north-east of the Terra Cimmeria - Sirenum province, in the southern hemisphere of Mars. Moreover, they infer that the NWA 7034 breccia belongs to the ejecta deposits of the Khujirt crater that date back to 1.5 Ga, and it was ejected because of the formation of the Karratha crater at 5-10 Ma. The authors then discuss the implications of this analysis for our understanding the early crustal record of Mars.

My main expertise does not lie in the field of crater analysis and, as such, I cannot provide an insightful review for this part of the study, and I imagine that the editor will seek advice from an additional referee on this topic. However, from a geological perspective and based on the thermal history of these meteorite breccias, the proposed launching site and history makes sense.

The main weakness of this paper lies in the last section, which discusses the implications for early crustal evolution on Mars. There are a number of statements in this section that are, in my view, in direct conflict with the data and interpretation reported in the recent Costa et al. (2020) paper. It is clear that the authors have not fully appreciated important aspects of this paper such as, for example, the composition of the primordial crust based on the U-Pb ages and Hf isotope compositions of ancient zircons. Although Bouvier et al. (2018) have indeed suggested that the composition of the primordial crust was andesitic as pointed out in the manuscript under evaluation, a more exhaustive zircon dataset reported in Costa et al. (2020) indicate that this may not be the case. Thus, the inference made in the manuscript with respect the existence of primary and secondary crusts may not be relevant. In sum, while I believe that this paper is interesting and important, there are serious shortcomings in the discussion that need to be addressed. I outlined these as well as other points in the detailed comments below, which are meant to be constructive and improve the paper - I hope that these will be helpful to the authors. Note that the comments are listed in order of appearance in the text.

Detailed comments

1-Around line 52 of the introduction, the authors state "...containing a variety of igneous, sedimentary, and impact melt clasts, 52 including the most evolved and oldest igneous clasts (4.47 - 4.48 Ga old1-9, in grey in Fig. 1)". The referencing is a bit odd here. First, they

reference Kruijer et al (2017), which does not have anything to do with Mars, clasts from NWA 7034 or zircon for that matter and, as such, should be removed. One of the oldest igneous clasts is the C27 basaltic clast reported by Costa et al. (2020), which is interpreted to have a crystallization age of 4443.6 ± 1.2 Ma based on zircons extracted from this clast. Thus, citing Costa et al. is appropriate here. They also include a reference to an unpublished abstract (ref. 7), which should be removed as pointed out in comment 4. Finally, the authors point out that the clasts are interpreted to be the product of remelting of an older crust by impacts. They should include a citation to the recent paper by Deng et al. (2020, Science Advances 6, eabc4941), which provides evidence for early melting of the clasts (including C27) by impacts.

2-The authors state in the last part of the introduction that knowledge of the source region of the NWA 7034 breccia and its pairs will “provide clues into the presence or absence of a local or global magma ocean, the conditions of the primitive crust extraction, and possibly the origin and the timing of the hemispheric dichotomy”. I think this is stretching it a bit as the authors cannot unequivocally speak towards the absence of presence of a magma ocean or, for that matter, the conditions of the primitive crust extraction based on their analysis. This should be toned down accordingly.

3-Figure 1 is difficult to read, especially for non-expert as most people would not know the difference between a U-Pb and Pb-Pb age. Perhaps this could be explained in the caption. There are also some confusing information. For example, the authors quote that the age of $4,428 \pm 25$ Ga from ref. 8 (Humayun et al., 2013) as being a Pb-Pb age. However, this is a concordia age and, as such, should be quoted as a U-Pb age. Perhaps some of the references have been mixed? Moreover, it does not seem that the authors have reported the ages of the ancient zircons from Costa et al. (2020) in this figure. Although they do report the young zircon population, the ancient ones appear to be missing. They should be included here as they represent the most exhaustive dataset of zircon ages from Mars (51 zircons and 2 baddeleyites).

4-When quoting zircon ages, the authors refer to Yin et al. (2014) (reference #7). This is a conference abstract and, as such, it not a refereed publication. Since it is not possible to evaluate the accuracy of the data based on the lack of details, this reference should be omitted. There are plenty of published high precision U-Pb and Pb-Pb ages that can be quoted.

5-Around line 91, the authors state that: “...(2) the ages of the oldest zircons found in the breccia, 7, 8, 11...” They do not refer to the Costa et al. (2020) paper, which in fact reports the oldest age for a zircon from these meteorites with an $207\text{Pb}/206\text{Pb}$ age of 4485.5 ± 2.2 Ma. It is also unclear to me why the authors refer to the Bellucci et al. paper here - this paper reports Pb isotopic composition of various Martian meteorites and not zircon ages. It is fine to cite this paper for the concept of an ancient crust but then the statement made by the authors must be modified.

6-At line 170, the authors state “...the clasts, including monzonitic and noritic melt rocks containing the concordant 4.4 Ga zircons⁸, represent the bedrock of the Khujirt crater...”

It is not clear to me why the bedrock should exclusively comprise evolved lithologies as suggested by the authors. As pointed out earlier, the most precise age for a single clast comes from the Costa et al. (2020), which show that a clast of basaltic composition contains concordant igneous zircons that define a $^{207}\text{Pb}/^{206}\text{Pb}$ age of 4443.6 ± 1.2 Ma. Thus, there is no reason that the bedrock is exclusively made of evolved lithologies - basaltic clasts do contain concordant zircons and this type of clasts is one of the most abundant clast type in NWA 7034.

7-At line 202, the authors state that: "The U-Pb isotopic compositions in monzonitic clasts in the meteorite suggest the existence of an enriched and differentiated crust on Mars 1-5, 46 that was extracted before 4.47 Ga and not subsequently recycled." This is not really accurate as the timing of primordial crust extraction is based on the Hf isotope composition of ancient zircons reported in both Bouvier et al. and Costa et al. and is 4.547 not 4.47 Ga. Moreover, it is not clear what is meant here with "enriched" - isotopically enriched? They should be more precise here.

8-At line 207, the authors discuss that the results of Bouvier et al. highlighting that the primordial crust may have been of andesitic composition and extracted before 4.547 Ga. However, this has been revised in Costa et al. based on a much larger zircon dataset. This is based on the fact that the primordial crust may have experience fractional crystallization of a Hf-bearing phase such as zircon and/or baddeleyite. Thus, the inferred Lu/Hf ratio from the zircon dataset may not hold compositional information. This point was made clearly in the section entitled "Formation Timescale and Reworking of the Primordial Martian Crust" in Costa et al. Thus, the authors need to revise this part of the discussion.

9-At line 209, it is stated that: "Since cooling of a global magma ocean on Mars would lead to the extraction of a primordial basaltic crust 42-45, 49." A number of papers by Lindy Elkins-Tanton have showed that it is possible to generate andesite compositions from the crystallization of a magma ocean - these are cited in the Bouvier et al. paper.

10-I am not too enthusiastic about the section regarding the lack of magma ocean on Mars, which is based on the inference made by Humayun et al. (2013). At that time, it was inferred that the so-called enriched reservoir responsible for the enriched shergottites was the ancient crust. Since then, a number of papers (i.e. Armytage et al. 2018, EPSL) have showed that the crust cannot account for the enrichment observed in shergottites, requiring a mantle origin for this compositional endmember. Moreover, as pointed out by Costa et al., the existence of a primitive "chondritic" reservoir in the deep martian is consistent with rapid initiation of solid-state convection of the martian mantle following magma ocean crystallization.

Reviewer #2 (Remarks to the Author):

This study is noteworthy in that it establishes an ejection site on Mars for the unique

martian meteorite NWA 7034 and its paired stones. It builds on previous work which have located the source region in the Noachian Highlands and combines it with previous work on possible ejection crater selection. An additional result from this study is that it puts constraints on the interpretations of possible heat sources for the later reheating event seen in multiple chronometers in clasts in this meteorite sample. While the methodology seems largely sound, there are a couple of aspects with communicating the criteria used to arrive at the unique solution, which require a bit more explanation and critical evaluation (see specific comments below). Overall, this is a sound piece of research which only requires minor revisions prior to publication.

Specific comments:

The authors systematically downselect possible crater options to arrive at a unique solution for NWA7034. However, there are a couple of aspects where the selection lacks explanation. First, the initial shortlisting of nineteen possible craters (L75-L85) mentions that these 19 constitute the “the complete crater population > 7 km in diameter formed on Mars over the last ~10 Ma, potentially responsible for the ejection of martian meteorites”. It is necessary to look at the authors’ recent paper (ref 17) to understand the size significance with respect to source craters for meteorites. An additional sentence of explanation here or in the Methods section would strengthen the argument that is being put forth. Second, L159 mentions the “five criteria” used to locate the crater source. It is unclear precisely which five are being referenced, as depending on how one reads L160-161 or Extended Table 1 there are either four or seven (and it depends whether the initial selection using the CDA is included as well, and whether K and Th are counted as two separate criteria). Having a statement at the beginning of the manuscript specifying which criteria are being used would be ideal.

There are obviously challenges associated with correlating spacecraft data and laboratory data. In addition to a bit more clarity as to the number of criteria used, some evaluation of which criteria are given the most weight would also be desirable particularly in terms of increasing the applicability of this approach for other meteorites in the future. It appears that the order was size of crater > lithology > chemistry > age, but a bit more discussion of how that order of selection was chosen would be good. It is not clear how useful the magnetization data was for example. The selection seemed largely based on geology (admittedly magnetization is not independent from lithology) and chronology.

Reviewer #3 (Remarks to the Author):

This paper provides important, well documented results. I recommend its publication.

In an attempt to follow the website recommendations for the I offer the following:

1. Key results- locating the probable Martian source of NWA 7034 allows more confidence in leveraging conclusions from this meteorite to Mars itself.
2. Validity - a number of lines of argument were offered. To the extent that I am able to

evaluate them they are valid.

3. Significance. The range of analytical techniques that have been applied to this meteorite will not be matched by analysis of real Martian samples for long time.

4. Data and Methodology - The authors draw on a variety of data sources for which results are available for the global Martian surface and which can be compared to the meteoritical results. For those areas where I have some expertise (chronology and geochemistry), the comparisons they make are valid.

5. Analytical approach - Analytical are compiled from the literature with an implicit statistical approach to comparisons of Mars and meteorite. The conclusions draw heavily on a previously published paper by the senior author which I did not review before making these comments.

6. Improvements - I have no suggestions for improvements.

7. Clarity - The paper is clearly written

8. References - To the best of my knowledge they are complete and appropriate.

9. Reviewer expertise - I claim some expertise in Martian meteorite analysis and have always attempted to connect what we learn from Martian meteorites to observations about Mars. I have gained some knowledge of Martian global characteristics from following the literature for about four decades.

10. - Other - I noticed only one or two typographical errors. One is at line 504. The illustrations are informative and well done. I am not a fan of the modern approach of having short papers with extensive Appendices, but it is a necessary evil. The extended data tables lend credibility to results reported in the main text.

The authors are to be congratulated for the manner in which they have brought diverse results together to present in a credible manner the probable origin of this very complex meteorite from the equally complex Martian surface.

Reviewer #4 (Remarks to the Author):

If I correctly understood the sequence of the events suggested by the authors (from the latest to the earliest):

a) NWA7034 ejection during the formation of a relatively young (5-10 Ma) 10-km-diameter Karratha crater.

b) The target area of this crater, as suggested by the authors, is covered by at least 60 m of ejecta from an older (1.25 - 1.87 Ga) Khujirt crater with a diameter of 40 km.

c) Khujirt crater excavated the primordial Martian crust and part of this crust was deposited within the area of a future Karratha crater.

The authors correctly estimated Khujirt ejecta thickness at the Karratha site (~ 55 km between centers of two craters). However, I cannot agree that Martian meteorites are excavated from a depth of 100 m from a 10-km-diameter crater (with a reference to my paper). The projectile size for this crater should be ~350 m, and the excavation depth of escaping ejecta cannot exceed 0.15 of the projectile radii, i.e., 50 m or, if substantial shock metamorphic features are not presented - from a shallower depth of 0.05 of the projectile radii, i.e.,

from 17 m as maximum (Artemieva and Ivanov 2004). However, this lower value of excavation depth does not contradict the authors' main idea (and even allows smaller thickness of Khujirt ejecta).

Modeling (iSALE-2D) efforts can be subdivided into two parts:

1. Modeling of the Khujirt crater to define temperature of ejected materials which are deposited at a distance of 55 km (future Karratha) crater.
2. Modeling additional heating of these materials during their deposition on the surface.

The first part may be OK, but need some work to be done. Temperature of 500° C looks like an average temperature in a hospital. Ejected materials at any velocities are subjected to a variety of shock pressures and, hence, temperatures. In other words, at any distance from the parent crater ejecta are a mixture of melt, highly shocked, and unshocked materials. To figure out the proportion between these materials, tracer particles are usually used. Then all tracers ejected with certain velocities (see below) should be analyzed from the viewpoint of their maximum shock compression (and hence, temperature). Average T is not interesting at all, but the range should be defined (and better - the proportion between melts, solids, etc). In addition, there are two temperatures during any impact event - maximum temperature during shock compression (quite high, but very short pulse of heating) and post-shock temperature (usually much lower but lasting for a much longer time). What temperature is shown in the Figure? The figure looks strange to me - why the ejecta curtain splits into two "branches", what are all these black dots? I suspect that those are artefacts due to impossibility to properly resolve high-velocity ejecta. Fortunately, it is not necessary, as ejecta at the Karratha site are not high-velocity ejecta (see next paragraph). The main point of part 1 - temperature has to be defined correctly with clear explanations - is it the highest temperature (related to maximum shock compression) or post-shock temperature.

In the second part (Fig. 6b) the strangest thing is the value of ejection velocity of 2 km/s allowing materials from the Khujirt crater to land at the Karratha site. The distance between crater centers is 55 km (correctly shown in ED Fig. 6a. Taking into account Mars gravity of 3.7 m/s², the ejection velocity should be at most 450 m/s, not 2 km/s. If the ejection velocity is 2 km/s, fragments are deposited at distances > 1000 km. Thus, Part 2 of modelling efforts have to be re-done or excluded. It is clear that at substantially lower velocity fragments are not re-heated upon the impact. Thus, all the observed chemical changes in NWA7034 probably take place during the ejection, not deposition. And, if part 1 is fulfilled correctly then certainly the authors find a fraction of ejecta with suitable shock conditions to prove their findings.

Natalia Artemieva

Planetary Science Institute

Response to reviewer's comments

Reviewer #1 (Remarks to the Author):

Dear reviewer,

Thank you very much for your constructive feedback on our manuscript. We have considered all the points raised and we address them below. Responses to each of your comments are highlighted in blue. We believe that your comments were very helpful in improving the discussion and the implications for the geological context of the meteorite. Please see detailed responses below.

This paper reports an analysis of Mars's cratering record aimed at identifying the source region of the NWA 7034 martian meteorite and its pairs (i.e. NWA 7533). These meteorites contain minerals and fragments of ancient crust that date back to 4.5 Ga and, as such, provide insights into the earliest crustal evolution on Mars. Thus, a robust identification of the launching site of these samples is important as it may guide future robotic exploration campaigns that aim to sample the primordial crust of Mars.

The authors' analysis suggests that the NWA 7034 meteorite suite was ejected around 5-10 Ma from the north-east of the Terra Cimmeria – Sirenum province, in the southern hemisphere of Mars. Moreover, they infer that the NWA 7034 breccia belongs to the ejecta deposits of the Khujirt crater that date back to 1.5 Ga, and it was ejected because of the formation of the Karratha crater at 5-10 Ma. The authors then discuss the implications of this analysis for our understanding the early crustal record of Mars.

My main expertise does not lie in the field of crater analysis and, as such, I cannot provide an insightful review for this part of the study, and I imagine that the editor will seek advice from an additional referee on this topic. However, from a geological perspective and based on the thermal history of these meteorite breccias, the proposed launching site and history makes sense.

The main weakness of this paper lies in the last section, which discusses the implications for early crustal evolution on Mars. There are a number of statements in this section that are, in my view, in direct conflict with the data and interpretation reported in the recent Costa et al. (2020) paper. It is clear that the authors have not fully appreciated important aspects of this paper such as, for example, the composition of the primordial crust based on the U-Pb ages and Hf isotope compositions of ancient zircons. Although Bouvier et al. (2018) have indeed suggested that the composition of the primordial crust was andesitic as pointed out in the manuscript under evaluation, a more exhaustive zircon dataset reported in Costa et al. (2020) indicate that this may not be the case. Thus, the inference made in the manuscript with respect to the existence of primary and secondary crusts may not be relevant. In sum, while I believe that this paper is interesting and important, there are serious shortcomings in the discussion that need to be addressed. I outlined these as well as other points in the detailed comments below, which are meant to be constructive and improve the paper – I hope that these will be helpful to the authors. Note that the comments are listed in order of appearance in the text.

All the questions raised above were addressed separately below.

Detailed comments

1-Around line 52 of the introduction, the authors state "...containing a variety of igneous, sedimentary, and impact melt clasts, 52 including the most evolved and oldest igneous clasts (4.47 - 4.48 Ga old 1-9, in grey in Fig. 1)". The referencing is a bit odd here. First, they reference Kruijer et al (2017), which does not have anything to do with Mars, clasts from NWA 7034 or zircon for that matter and, as such, should be removed. One of the oldest igneous clasts is the C27 basaltic clast reported by Costa et al. (2020), which is interpreted to have a crystallization age of 4443.6 ± 1.2 Ma based on zircons extracted from this clast. Thus, citing Costa et al. is appropriate here. They also include a reference to an unpublished abstract (ref. 7), which should be removed as pointed out in comment 4. Finally, the authors point out that the clasts are interpreted to be the product of remelting of an older crust by impacts. They should include a citation to the recent paper by Deng et al. (2020, Science Advances 6, eabc4941), which provides evidence for early melting of the clasts (including C27) by impacts.

We agree that referencing Kruijer et al., 2017 is not appropriate in the context of our study and therefore has been removed in the revised version. We also removed the non-peer reviewed study by Yin et al. (2014) as suggested, and referenced Costa et al. (2020) when mentioning the age of the oldest clasts and zircons found in the meteorite. Figure 1 and references have been modified accordingly. Deng et al. study is also cited when mentioning the remelting of the crust by impacts.

2-The authors state in the last part of the introduction that knowledge of the source region of the NWA 7034 breccia and its pairs will "provide clues into the presence or absence of a local or global magma ocean, the conditions of the primitive crust extraction, and possibly the origin and the timing of the hemispheric dichotomy". I think this is stretching it a bit as the authors cannot unequivocally speak towards the absence of presence of a magma ocean or, for that matter, the conditions of the primitive crust extraction based on their analysis. This should be toned down accordingly.

We agree that as currently written, the end of the introduction oversold the implications of our study in relation with the magma ocean scenario. We rephrased L.65: "Knowing this source region would provide insights into early Mars geological history and crustal extraction^{2,3}."

3-Figure 1 is difficult to read, especially for non-expert as most people would not know the difference between a U-Pb and Pb-Pb age. Perhaps this could be explained in the caption. There are also some confusing information. For example, the authors quote that the age of $4,428 \pm 25$ Ga from ref. 8 (Humayun et al., 2013) as being a Pb-Pb age. However, this is a concordia age and, as such, should be quoted as a U-Pb age. Perhaps some of the references have been mixed? Moreover, it does not seem that the authors have reported the ages of the ancient zircons from Costa et al. (2020) in this figure. Although they do report the young zircon population, the ancient ones appear to be missing. They should be included here as they represent the most exhaustive dataset of zircon ages from Mars (51 zircons and 2 baddeleyites).

We appreciate the careful check on our referencing and ages details in the text and Figure 1. The age of 4.428 Ga from Humayun et al. study is now reported as a U-Pb age in Fig. 1. We also incorporated Costa et al. (2020) old zircon population and baddeleyite ages. Finally, we

compiled ages and dates we used to produce Fig.1 in Supplementary Table 1 (not claiming completeness). We also updated the figure caption to improve clarity: "Summary of NWA 7034 and paired stone radiometric ages, and chronology of major events experienced by the breccia. Dates from each study are reported in Supplementary Table 1. The ejection event is constrained from $^{22}\text{Ne}/^{21}\text{Ne}$ cosmic ray exposure ages^{14,15}. Whole rock (w.r), zircon and or baddeleyite (zr.b), apatite (ap), and augite (aug) on which ages have measured are also mentioned. The different chronometers used in these studies are reported (Sm-Nd, Pb-Pb, U-Pb, K-Ar, U-Th/He, U-Th-Sm/He). Note that green, red and blue boxes correspond to resetting ages of the noted chronometer. Reset ages in green are widely interpreted as the disruption induced by an impact-derived heating event that has excavated the oldest components of the breccia ~1.5 ago^{9,13,23,26}, although its precise age is still unconstrained due to the wide range of isotopic dates reported in the literature."

4-When quoting zircon ages, the authors refer to Yin et al. (2014) (reference #7). This is a conference abstract and, as such, it not a refereed publication. Since it is not possible to evaluate the accuracy of the data based on the lack of details, this reference should be omitted. There are plenty of published high precision U-Pb and Pb-Pb ages that can be quoted.

Thanks for pointing that out. As mentioned above, this reference has been removed. Regarding the addition of other references of U-Pb and Pb-Pb ages, we think that the references already cited in the manuscript give a satisfying overview of the old zircon population ages.

5-Around line 91, the authors state that: "... (2) the ages of the oldest zircons found in the breccia 1,7,8,11..." They do not refer to the Costa et al. (2020) paper, which in fact reports the oldest age for a zircon from these meteorites with an $^{207}\text{Pb}/^{206}\text{Pb}$ age of 4485.5 ± 2.2 Ma. It is also unclear to me why the authors refer to the Bellucci et al. paper here – this paper reports Pb isotopic composition of various Martian meteorites and not zircon ages. It is fine to cite this paper for the concept of an ancient crust but then the statement made by the authors must be modified.

Thank you again for pointing that out. References we used in this sentence are now Bouvier et al., 2019, Costa et al., 2020, Humayun et al., 2013, McCubbin et al., 2016 and Hu et al., 2019.

6-At line 170, the authors state "...the clasts, including monzonitic and noritic melt rocks containing the concordant 4.4 Ga zircons⁸, represent the bedrock of the Khujirt crater..." It is not clear to me why the bedrock should exclusively comprise evolved lithologies as suggested by the authors. As pointed out earlier, the most precise age for a single clast comes from the Costa et al. (2020), which show that a clast of basaltic composition contains concordant igneous zircons that define a $^{207}\text{Pb}/^{206}\text{Pb}$ age of 4443.6 ± 1.2 Ma. Thus, there is no reason that the bedrock is exclusively made of evolved lithologies – basaltic clasts do contain concordant zircons and this type of clasts is one of the most abundant clast type in NWA 7034.

We agree that the word "represent" in this sentence is confusing and that the data does not support a bedrock exclusively composed of evolved lithologies. We modified this sentence and now cite Costa et al. (2020), L. 176: "The bedrock of the Khujirt crater, located in the north-east of the Terra Cimmeria – Sirenum region (here and after noted TCTS) is believed to be composed of basaltic and more evolved lithologies such as those represented by the monzonitic and noritic clasts, containing the concordant 4.4 Ga zircons^{1-5,8-10}."

7-At line 202, the authors state that: "The U-Pb isotopic compositions in monzonitic clasts in the meteorite suggest the existence of an enriched and differentiated crust on Mars^{1-5,46} that

was extracted before 4.47 Ga and not subsequently recycled¹.” This is not really accurate as the timing of primordial crust extraction is based on the Hf isotope composition of ancient zircons reported in both Bouvier et al. and Costa et al. and is 4.547 not 4.47 Ga. Moreover, it is not clear what is meant here with “enriched” – isotopically enriched? They should be more precise here.

Thank you, we fixed the error on the age L. 202. We also clarified the type of enrichment we meant. This sentence reads now: “The U-Pb and Pb-Pb isotopic compositions in monzonitic clasts in the meteorite suggest the existence of an isotopically enriched (relative to the martian mantle) and differentiated crust on Mars^{1-5,11,50} that was extracted before 4.547 Ga¹.”

8-At line 207, the authors discuss that the results of Bouvier et al. highlighting that the primordial crust may have been of andesitic composition and extracted before 4.547 Ga. However, this has been revised in Costa et al. based on a much larger zircon dataset. This is based on the fact that the primordial crust may have experience fractional crystallization of a Hf-bearing phase such as zircon and/or baddeleyite. Thus, the inferred Lu/Hf ratio from the zircon dataset may not hold compositional information. This point was made clearly in the section entitled “Formation Timescale and Reworking of the Primordial Martian Crust” in Costa et al. Thus, the authors need to revise this part of the discussion.

Costa et al. (2020) present additional Lu/Hf ratio of zircon that correspond to an enriched crustal component, and the presence of one zircon with a low ratio in one basaltic clast led to the interpretation of low Lu/Hf related to fractional crystallization of zircon. Although totally possible for the basaltic clast, evolved clasts in the breccia along with evolved igneous rocks analysed on Mars point out an evolved crustal component very early in Mars history. The inversion of the field of gravity of Mars, constrained by petrological data (expected density of Martian basalts) (Baratoux et al., 2014) indicate that the bulk composition of the martian crust is not basaltic, whereas the surface composition is essentially basaltic. The solution of this equation resides in the presence of evolved (less dense than basalt) components, likely buried underneath late basaltic material. In addition, seismic data suggest a vp/vs ratio corresponding to that traveling through basaltic to andesitic materials (Deng and Levander, 2020) as well as a crust density <3100g.cm⁻³ (Knapmeyer-Endrun et al., 2021), so lower than if only composed by basaltic rocks). Overall, both geochemical and geophysical datasets support the presence of ancient evolved crustal components, likely andesitic, which could be remnants of either a primordial (extracted from the mantle) or a secondary crust (involving re-melting of the primordial crust).

We discussed and strengthened this in the text, L. 225-236: “The initial ϵ_{Hf} value of these old zircons and models of early magma ocean crystallisation⁵¹⁻⁵³ imply an andesitic composition for the early martian crust^{1,5,53}, although the relationship between U-Pb ages and ϵ_{Hf} of the oldest zircons suggest that they crystallized from low ¹⁷⁶Lu/¹⁷⁷Hf magmas potentially of basaltic affinity⁵. Such a crust was then reworked 100 Ma later by impacts^{9,10}, producing the melts from which the old zircons crystallized. The analyses of > 3.8 Ga evolved rocks in Gale crater⁴⁶, the inversion of the martian gravity field (constrained by petrological data that support the existence of light evolved crustal components -less dense than basalt- in the southern highlands)⁵⁴ and, finally, seismic data from the *Insight* mission (indicating that the rate of P wave against S wave is compatible with basaltic to andesitic crustal materials⁵⁵ and suggesting that the crust density is <3,100 g.cm⁻³, so lower if only composed by basaltic rocks⁵⁶), all point out to the presence of highly ancient evolved crustal components in TCTS.”

9-At line 209, it is stated that: “Since cooling of a global magma ocean on Mars would lead to the extraction of a primordial basaltic crust^{42-45,49}.” A number of papers by Lindy Elkins-

Tanton have showed that it is possible to generate andesite compositions from the crystallization of a magma ocean – these are cited in the Bouvier et al. paper.

Elkins-Tanton studies indeed mainly mention a basaltic crust extracted from a magma ocean, except in one study (Elkins-Tanton et al., 2005) where the possibility of a basaltic to andesitic crust extraction is mentioned, based on an ancient version of the MELTS thermodynamical calculator (valid for low pressure). However, partial melting of Mars primitive mantle compositions cannot form andesitic compositions according to both partial melting experiments (Collinet et al., 2015) and adiabatic ascent pMELTS (valid for pressure up to 3 GPa) modelling of primitive mantle compositions (Payré et al., 2020).

We now discuss the possibility of crystallization from deep melts in the discussion as suggested by Elkins-Tanton et al. (2005), L. 237-241: “If andesitic in composition, the crust could either be secondary (reworking of the primordial crust) or primordial. If primordial, basaltic to andesitic melts that originated from a deep mantle source might have crystallized⁵⁷. However, isobaric partial melting experiments⁵⁸ and adiabatic ascent of primitive mantle compositions⁴⁸ argue against the formation of andesitic magmas under such a scenario.”.

10-I am not too enthusiastic about the section regarding the lack of magma ocean on Mars, which is based on the inference made by Humayun et al. (2013). At that time, it was inferred that the so-called enriched reservoir responsible for the enriched shergottites was the ancient crust. Since then, a number of papers (i.e. Arnytage et al. 2018, EPSL) have showed that the crust cannot account for the enrichment observed in shergottites, requiring a mantle origin for this compositional endmember. Moreover, as pointed out by Costa et al., the existence of a primitive “chondritic” reservoir in the deep martian is consistent with rapid initiation of solid-state convection of the martian mantle following magma ocean crystallization.

The paper does not rule out the existence of a magma ocean on Mars, but suggests that perhaps, it was not a global magma ocean as it is envisioned in a number of papers, including in Elkins-Tanton et al., 2005. We are discussing the two possibilities in the text. Concerning Humayun et al. (2013), they show that a low degree of partial melting of a primitive mantle composition leads to a similar REE pattern observed in ICM and CLIMR, both suggested to be able to provide information regarding the formation of the primary crust based on their geochemical compositions close to martian soils. This section has been modified accordingly.

Reviewer #2 (Remarks to the Author):

Dear reviewer,

Thank you very much for your constructive feedback on our manuscript. We have considered all the points raised and we address them below. Responses to each of your comments are highlighted in brown. We believe that your comments were very helpful in improving the readability and the presentation of the criteria we used to pinpoint the crater source of the meteorite. Please see detailed responses below.

This study is noteworthy in that it establishes an ejection site on Mars for the unique martian meteorite NWA 7034 and its paired stones. It builds on previous work which have located the source region in the Noachian Highlands and combines it with previous work on possible ejection crater selection. An additional result from this study is that it puts constraints on the interpretations of possible heat sources for the later reheating event seen in multiple chronometers in clasts in this meteorite sample. While the methodology seems largely sound, there are a couple of aspects with communicating the criteria used to arrive at the unique solution, which require a bit more explanation and critical evaluation (see specific comments below). Overall, this is a sound piece of research which only requires minor revisions prior to publication.

Specific comments:

The authors systematically downselect possible crater options to arrive at a unique solution for NWA7034. However, there are a couple of aspects where the selection lacks explanation. First, the initial shortlisting of nineteen possible craters (L75-L85) mentions that these 19 constitute the “the complete crater population > 7 km in diameter formed on Mars over the last ~10 Ma, potentially responsible for the ejection of martian meteorites”. It is necessary to look at the authors’ recent paper (ref 17) to understand the size significance with respect to source craters for meteorites. An additional sentence of explanation here or in the Methods section would strengthen the argument that is being put forth.

The brevity of this paragraph was a consequence of the journal format. We agree that the identification of these 19 candidates constitutes an important step in the present study and needs to be developed. This is now done in the main text where further details linked to our previous study are provided along the original paragraph you quoted, focusing on the completeness of this young crater population.

L. 81-97: “Following a hypervelocity impact, ejecta materials faster than the escape velocity (5 km/s¹⁷) may get through the martian atmosphere and continue their course into interplanetary space to become martian meteorites. Slower debris fall back on the surface in a radial pattern or ray around the primary crater, forming secondary craters. Due to erosion conditions on the surface, the presence of 100 meter-size secondaries attests to the freshness of their associated primary craters¹⁸. Using the size and spatial distribution of more than 90 million impact craters >50 m detected using a Crater Detection Algorithm (CDA)¹⁸⁻²⁰ on the whole surface of Mars from the global Context Camera (CTX) mosaic²¹, a previous work¹⁸ identified ray systems of secondary craters < 150 m associated with 19 large primary craters. For each of them, a formation model age was measured using small craters superposed on their ejecta blanket, and 18 were found younger than 10 Ma old. The analysis of the size frequency distribution of these 18 young crater candidates revealed that those larger than 7 km (i.e. 17 out of 18) align with the predicted number and size of craters accumulated on the whole surface of Mars over the last 8.2±2 Ma²². Hence, those impact craters were found to

constitute the complete crater population > 7 km in diameter formed on Mars over the last ~10 Ma, potentially responsible for the ejection of martian meteorites¹⁸. One of these craters, Tooting, has already been recognized as the most likely ejection site of the depleted olivine-phyric shergottites launched 1.1 Ma ago, located on the Tharsis volcanic province¹⁸.”

Second, L159 mentions the “five criteria” used to locate the crater source. It is unclear precisely which five are being referenced, as depending on how one reads L160-161 or Extended Table 1 there are either four or seven (and it depends whether the initial selection using the CDA is included as well, and whether K and Th are counted as two separate criteria). Having a statement at the beginning of the manuscript specifying which criteria are being used would be ideal.

The 19 candidates pinpointed in our previous study were implicitly included in the five criteria mentioned L.159. We agree that this induces confusion. Our criteria are now listed in the section “Constraints on the meteorite launch site” to clarify this. Note that while the presence of secondary crater rays associated with large craters is a criterion in itself, we chose to discard this aspect from the list to improve the flow of the paragraph.

L. 103-109: “In this study, we search for the most likely site of ejection of the regolith breccia by using four criteria based on its geochemical and geophysical properties as well as its geochronological records (Methods), which we compare to potential sites based on their known properties and geological context: (1) updated maps of magnetic field intensity and remanent magnetization at the surface²⁷ (Fig. 2b and 2c); (2) elemental Th and K concentrations^{28,29} (Figs. 2d and 2e) of the areas surrounding each crater candidate; (3) superposition on a Noachian geological unit³⁰ and (4) connection with material from an Early Amazonian impact.”

There are obviously challenges associated with correlating orbital data and laboratory data. In addition to a bit more clarity as to the number of criteria used, some evaluation of which criteria are given the most weight would also be desirable particularly in terms of increasing the applicability of this approach for other meteorites in the future . It appears that the order was size of crater > lithology > chemistry > age, but a bit more discussion of how that order of selection was chosen would be good. It is not clear how useful the magnetization data was for example. The selection seemed largely based on geology (admittedly magnetization is not independent from lithology) and chronology.

That is a good point we discussed in the preliminary phase of this study. However, we did not assign any weight to the criteria we used. Our aim was to test each of them with respect to all candidates and discuss any mismatch. The order with which we present each of the criteria in the manuscript constitutes the easiest way to downselect crater candidates. This is because the two first (superposition on a Noachian unit and on an Early impact crater material) are binaries (each candidate is or is not superposed on a Noachian geological unit and/or on an Amazonian impact crater material), whereas the two last (elemental abundances and magnetic signatures) are quantitative criteria whose face values are relevant in the context of our study only if compared between each candidate: as you mentioned in your comment, the correspondence between orbital and laboratory data is challenging. The magnetic data and elemental abundances were given the same weight in the downselection process. Finally, the model age derivation of the craters on which Gasa and Karratha are superposed was used only to quantify the age of the two impact craters material on which Gasa and Karratha craters are superposed. Extended Data Figure 3 summarizes each criterion where these two craters appear as the most convincing candidates, if all criteria are taken into account. The methodology and arguments we use here are unique, specific to this study as they are directly dependant on the meteorite characteristics. Therefore, this is not transposable to any other type of martian meteorite.

As the criteria used to pinpoint the crater source of NWA 7034 are now explicitly stated in the revised manuscript (see response above), and because of the reasons mentioned above, we do not think that further discussions on criterion ranking is necessary to clarify the manuscript or support our conclusion.

Reviewer #3 (Remarks to the Author):

Dear reviewer,

Thank you very much for your positive feedback on our manuscript.

This paper provides important, well documented results. I recommend its publication.

In an attempt to follow the website recommendations for the I offer the following:

1. Key results- locating the probable Martian source of NWA 7034 allows more confidence in leveraging conclusions from this meteorite to Mars itself.
2. Validity - a number of lines of argument were offered. To the extent that I am able to evaluate them they are valid.
3. Significance. The range of analytical techniques that have been applied to this meteorite will not be matched by analysis of real Martian samples for long time.
4. Data and Methodology - The authors draw on a variety of data sources for which results are available for the global Martian surface and which can be compared to the meteoritical results. For those areas where I have some expertise (chronology and geochemistry), the comparisons they make are valid.
5. Analytical approach - Analytical are compiled from the literature with an implicit statistical approach to comparisons of Mars and meteorite. The conclusions draw heavily on a previously published paper by the senior author which I did not review before making these comments.
6. Improvements - I have no suggestions for improvements.
7. Clarity - The paper is clearly written
8. References - To the best of my knowledge they are complete and appropriate.
9. Reviewer expertise - I claim some expertise in Martian meteorite analysis and have always attempted to connect what we learn from Martian meteorites to observations about Mars. I have gained some knowledge of Martian global characteristics from following the literature for about four decades.
10. - Other - I noticed only one or two typographical errors. One is at line 504. The illustrations are informative and well done. I am not a fan of the modern approach of having short papers with extensive Appendices, but it is a necessary evil. The extended data tables lend credibility to results reported in the main text.

Thank you – typographical errors are corrected.

The authors are to be congratulated for the manner in which they have brought diverse results together to present in a credible manner the probable origin of this very complex meteorite from the equally complex Martian surface.

Reviewer #4 (Remarks to the Author):

Dear Dr Artemieva,

Thank you very much for your constructive feedback on our manuscript. We have considered all the points raised and we address them below. Responses to each of your comment are highlighted in green. We believe that your comments were very helpful in improving the accuracy of the analysis we used to pinpoint the crater source of the meteorite. Please see detailed responses below.

If I correctly understood the sequence of the events suggested by the authors (from the latest to the earliest):

- a) NWA7034 ejection during the formation of a relatively young (5-10 Ma) 10-km-diameter Karratha crater.
- b) The target area of this crater, as suggested by the authors, is covered by at least 60 m of ejecta from an older (1.25 – 1.87 Ga) Khujirt crater with a diameter of 40 km.
- c) Khujirt crater excavated the primordial Martian crust and part of this crust was deposited within the area of a future Karratha crater.

The authors correctly estimated Khujirt ejecta thickness at the Karratha site (~ 55 km between centers of two craters). However, I cannot agree that Martian meteorites are excavated from a depth of 100 m from a 10-km-diameter crater (with a reference to my paper). The projectile size for this crater should be ~350 m, and the excavation depth of escaping ejecta cannot exceed 0.15 of the projectile radii, i.e., 50 m or, if substantial shock metamorphic features are not presented – from a shallower depth of 0.05 of the projectile radii, i.e., from 17 m as maximum (Artemieva and Ivanov 2004). However, this lower value of excavation depth does not contradict the authors' main idea (and even allows smaller thickness of Khujirt ejecta).

Thank you for your comment. We fixed the maximum excavation depth of escaping ejecta following your recommendation.

Modeling (iSALE-2D) efforts can be subdivided into two parts:

1. Modeling of the Khujirt crater to define temperature of ejected materials which are deposited at a distance of 55 km (future Karratha) crater.
2. Modeling additional heating of these materials during their deposition on the surface.

The first part may be OK, but need some work to be done. Temperature of 500°C looks like an average temperature in a hospital. Ejected materials at any velocities are subjected to a variety of shock pressures and, hence, temperatures. In other words, at any distance from the parent crater ejecta are a mixture of melt, highly shocked, and unshocked materials. To figure out the proportion between these materials, tracer particles are usually used. Then all tracers ejected with certain velocities (see below) should be analyzed from the viewpoint of their maximum shock compression (and hence, temperature). Average T is not interesting at all, but the range should be defined (and better – the proportion between melts, solids, etc). In addition, there are two temperatures during any impact event – maximum temperature during shock compression (quite high, but very short pulse of heating) and post-shock temperature (usually much lower but lasting for a much longer time). What temperature is shown in the

Figure? The figure looks strange to me - why the ejecta curtain splits into two “branches”, what are all these black dots? I suspect that those are artefacts due to impossibility to properly resolve high-velocity ejecta. Fortunately, it is not necessary, as ejecta at the Karratha site are not high-velocity ejecta (see next paragraph). The main point of part 1 – temperature has to be defined correctly with clear explanations - is it the highest temperature (related to maximum shock compression) or post-shock temperature (after the shock wave passes).

Simulations have been updated to include both shock and post shock temperatures (Extended Data Figure 6.a and b). We computed the distribution of shock temperatures in numerical cells in the proximal ejecta, at 55 km from the crater centre, where Khujirt crater was formed (Extended Data Figure 7.a and b). This shows that there are ejecta that satisfy the required temperature conditions to account for resetting ages measured in the breccia (>500°C).

In the original simulation, the ejecta curtain splits due to issues involving the basalt ANEOS that was developed in Pierazzo et al., 2005. To avoid the ejecta splitting effect, this simulation was re-done using the Tillotson equation of state for basalt (Benz and Asphaug, 1999). All other impact parameters are kept the same.

E. Pierazzo, N. A. Artemieva, B. A. Ivanov, Starting conditions for hydrothermal systems underneath Martian craters: Hydrocode modeling. In Large Meteorite Impacts III, T. Kenkmann, F. Hörz, A. Deutsch, Eds. (Geological Society of America, Boulder, CO, 2005), pp. 443–457.

Benz, W. & Asphaug, E. Catastrophic disruptions revisited. Icarus 142(1), 5-20 (1999). 10.1006/icar.1999.6204

In the second part (Fig. 6b) the strangest thing is the value of ejection velocity of 2 km/s allowing materials from the Khujirt crater to land at the Karratha site. The distance between crater centers is 55 km (correctly shown in ED Fig. 6a. Taking into account Mars gravity of 3.7 m/s², the ejection velocity should be at most 450 m/s, not 2 km/s. If the ejection velocity is 2 km/s, fragments are deposited at distances > 1000 km. Thus, Part 2 of modelling efforts have to be re-done or excluded. It is clear that at substantially lower velocity fragments are not reheated upon the impact. Thus, all the observed chemical changes in NWA7034 probably take place during the ejection, not deposition. And, if part 1 is fulfilled correctly then certainly the authors find a fraction of ejecta with suitable shock conditions to prove their findings.

We agree that this value is significantly higher than the actual impact velocity of the landing ejecta blanket, therefore it has been removed to avoid further confusion. The updated simulation shown in new Extended Data Figures 6 and 7 demonstrates that there is a fraction of the ejecta that satisfies the temperature required to reset radiochronometers measured in the breccia. The text and methods have been updated accordingly.

Natalia Artemieva

Planetary Science Institute

REVIEWERS' COMMENTS

Reviewer #1 (Remarks to the Author):

The authors have addressed my comments in a satisfactory manner and I am happy to recommend the paper for publication in Nature Communications.

Reviewer #2 (Remarks to the Author):

In their response to reviewers, the authors addressed all the concerns I had with respect to the initial submission. I recommended the publication of this noteworthy study.

Reviewer #4 (Remarks to the Author):

A few minor comments:

1. Extended Data Fig.6 does not illustrate ejecta blanket temperature as in reality it is very low on average and, in addition, iSALE (as any other Eulerian code) tends to show lower temperatures than the real ones due to numerical diffusion. I would suggest to rename the figure: illustration of crater formation and ejection of rocks to be deposited near the Karratha site.
2. Shock T is much higher than post-shock temperature but lasts, maybe, a fraction of a second or less. Can the authors speculate how long it takes to reset Ar-Ar and U-Pb chronometers in the ejected rocks? Post-shock temperature remains above average for a much longer time. Do the models (tracers) show some minor fraction of ejected materials with substantially elevated (500-800 C) post-shock T? It can be shown in Fig. 7 along with shock temperatures. I am not sure, but it is quite possible that even tracers are not able to reveal correct post-shock temperatures (as tracers could be easily lost during the ejection). I may recommend to use ANEOS directly to calculate post-shock conditions from the known shock conditions.
3. Line 575: Furthermore, any macro voids within the falling ejecta could elevate the temperature in the falling ejecta by up to 4 times [78]. I would say that it is a highly questionable statement – ejecta near the Karratha site are deposited at very low velocities (the authors can mention the proper range) whereas reference [78] deals with much higher velocities. Temperatures indeed could be higher during the Khujirt crater formation if the target has some porosity.
4. and the next statement – excavation depth of a 40 km diameter crater is certainly smaller than 10 km (I would expect not more than 4-5 km, the rule of a thumb is 1/10 of the transient crater diameter).

Reviewer #4 (Remarks to the Author):

A few minor comments:

1. Extended Data Fig.6 does not illustrate ejecta blanket temperature as in reality it is very low on average and, in addition, iSALE (as any other Eulerian code) tends to show lower temperatures than the real ones due to numerical diffusion. I would suggest to rename the figure: illustration of crater formation and ejection of rocks to be deposited near the Karratha site.

Thank you for this suggestion. We adapted the figure title as follow: "Illustration from iSALE 2D simulation of crater formation and rocks to be deposited following the Khujirt crater formation, near the Karratha site."

2. Shock T is much higher than post-shock temperature but lasts, maybe, a fraction of a second or less. Can the authors speculate how long it takes to reset Ar-Ar and U-Pb chronometers in the ejected rocks? Post-shock temperature remains above average for a much longer time. Do the models (tracers) show some minor fraction of ejected materials with substantially elevated (500-800 C) post-shock T? It can be shown in Fig. 7 along with shock temperatures. I am not sure, but it is quite possible that even tracers are not able to reveal correct post-shock temperatures (as tracers could be easily lost during the ejection). I may recommend to use ANEOS directly to calculate post-shock conditions from the known shock conditions.

The Ar-Ar and U-Pb chronometers resetting times are mainly function of the considered mineral, its size and temperature. The most detailed study we are aware of that treat this topic in the context of the meteorite is from MacArthur et al., 2019 (<https://doi.org/10.1016/j.gca.2018.11.026>). However, comparing in-situ micro-analysis with impact simulation may not be easily applicable, because of the unconstrained parameters that could affect the temperature behaviour (e.g. porosity, heterogeneity, sample size...). The purpose of the modelling we performed in this work is to test the possibility of ejecta experiencing post shock temperature higher than required for the chronometer resetting. We acknowledge that further work is needed to constrain post-shock conditions, but believe that the work presented here is sufficient to support our conclusions on the meteorite ejection site. We added a clarification in the text (L.402): "It is therefore difficult to estimate the duration of the shock temperatures necessary for chronometer resetting, as that would depend on a number of parameters such as mineral composition, sample/ejecta fragment size, porosity, heterogeneity, etc. Direct comparison between numerical modelling outcomes and laboratory measurements is not directly applicable, and merits further work."

3. Line 575: Furthermore, any macro voids within the falling ejecta could elevate the temperature in the falling ejecta by up to 4 times [78]. I would say that it is a highly questionable statement – ejecta near the Karratha site are deposited at very low velocities (the authors can mention the proper range) whereas reference [78] deals with much higher velocities. Temperatures indeed could be higher during the Khujirt crater formation if the target has some porosity.

We agree that this statement is questionable as the ejecta are deposited at about 350-450m/s (obtained from a ballistic trajectory of fragments ejected at 45° and deposited at 55km from the impact) and not a few km/s as used in ref [78]. We modified the sentence as follow: "Furthermore, any macro voids within the falling ejecta could significantly elevate the temperature in the falling ejecta⁷⁸".

4. and the next statement – excavation depth of a 40 km diameter crater is certainly smaller than 10 km (I would expect not more than 4-5 km, the rule of a thumb is 1/10 of the transient crater diameter).

This is right. We modified the sentence as follow: “Depth of origin for this ejecta material does not exceed 5 km.”